# Exploring barriers to accessing healthy diets among pregnant women living with HIV in the Njombe region, Tanzania: A qualitative study

Julieth Shine[1], Vera Kwara[2], Deborah Esau[2], Fatma Abdallah[1], Anna Zangira[1], Aika Lekey[1], Abela Twinomujuni[1], Maria Ngilisho[1], Zahara Amiri[1], Elizabeth Lyimo[1], Winfrida Akyoo[3], Germana H. Leyna[1,3], Ray M. Masumo[1]*

1 Tanzania Food and Nutrition Centre, Dar es Salaam, Tanzania, 2 The World Food Programme (WFP) Tanzania, Dar es Salaam, Tanzania, 3 Muhimbili University Health and Allied Sciences, MUHAS, Dar es Salaam, Tanzania

* rmasumo@yahoo.com

## Abstract

Proper nutrition is essential for women with HIV during pregnancy; however, information on how to access nutritious diets is scarce. This study, informed by the socio-ecological perspectives, examines the barriers to obtaining a healthy diet among pregnant women living with HIV in the Njombe region of Tanzania. A qualitative study employing the ethnography method using semi-structured and narrative interviews to gather information from key informant interviews, indepth interviews and focus group discussions engaged a diverse range of stakeholders. The analysis was done by MAXQDA software employing qualitative content analysis. Further, the thematic analysis was carried out by assigning data into relevant codes to generate categories based on study objectives. The study found inadequate nutritional knowledge among individuals and some healthcare providers in the Njombe region. Poor emotional and physical support from spouses and family members caused pregnant women living with HIV to shoulder an excessive household workload, leading to exhaustion and stress, which hindered their ability to practice healthy dietary behaviors. The level of alcohol consumption was high, posing a risk to their health and well-being. The study identified significant barriers at the individual, community, environmental, and organizational levels that prevent pregnant women living with HIV in the Njombe region of Tanzania from accessing healthy diets. Elevating nutritional awareness within these communities is essential for improving the knowledge, skills, and motivation of pregnant women, their partners, and the wider community to embrace healthy and nutritious dietary practices. While various obstacles to healthy diets may exist, motivation and intentionality in pursuing those dietary choices are equally important. Even in the face of challenges, individuals with a strong understanding of nutrition are more likely to discover alternative strategies to maintain healthy diets.

**Data availability statement:** Available here, URL: https://osf.io/nbxw9.

**Funding:** The author(s) received no specific funding for this work.

**Competing interests:** The authors have declared that no competing interests exist.

## Introduction

Diets during pregnancy in many low- and middle-income countries (LMICs), including Tanzania, are critically imbalanced in terms of essential macronutrients and micronutrients. These diets primarily rely on plant-based foods, which is inadequate for supporting the health needs of pregnant individuals [1,2]. The World Health Organization (WHO) defines a healthy diet during pregnancy as providing sufficient energy, protein, vitamins, and minerals through various foods, including green and orange vegetables, meat, fish, beans, nuts, dairy products, and fruits [1,2]. Poor nutrition during pregnancy is linked to one-fifth of maternal deaths and an increase in adverse pregnancy outcomes [3].

Proper nutrition is crucial for HIV-infected women during pregnancy, as they tend to gain less weight and face more micronutrient deficiencies compared to uninfected pregnant women [3]. Evidence suggests that HIV increases metabolic rate and elevates the caloric and nutrient requirements of individuals. Concurrently, HIV diminishes nutrient intake, impairs nutrient absorption, and adversely affects nutrient utilization. These factors may significantly contribute to malnutrition among people living with HIV(PLHIV), including pregnant women [3–5]. Moreover, HIV unequivocally elevates oxidative stress in individuals living with the virus, including pregnant women. This surge in oxidative stress leads to considerable cellular damage due to excessive free radicals and a marked reduction in immune function, severely undermining the body's ability to effectively combat diseases [4–6].

Opportunistic infections represent a significant challenge to food intake among individuals living with HIV, including pregnant women living with HIV [7]. Conditions such as oral thrush, esophageal candidiasis, and gastrointestinal complications can lead to painful swallowing, diminished appetite, and difficulties with food consumption. These infections can diminish appetite as eating becomes challenging and sore due to oropharyngeal infections [7]. Additionally, the medications taken by people living with HIV for HIV and opportunistic infections can interact with certain foods and nutrients. This interaction creates unique nutritional needs and dietary challenges for PLHIV, including pregnant women. Various studies have shown that Antiretroviral Therapies (ARVs) can impact nutrient absorption, metabolism, and distribution in the body. Moreover, ARVs may cause side effects that reduce food intake and absorption, leading to increased nutrient losses among PLHIV. As a result, their overall nutritional status can be negatively affected [8].

In addition to individual factors, various barriers impede access to healthy diets for pregnant women, particularly those who are HIV-infected [9,10]. Research has identified several significant obstacles to achieving an optimal maternal diet, including inadequate guidance from influential family members, economic limitations, inequitable distribution of food within households, food scarcity, limited exposure to nutrition counseling, entrenched gender norms, and insufficient knowledge regarding nutrition [9,10].

A study conducted in rural communities of Tigray in Northern Ethiopia has identified several barriers that hinder the uptake of nutrition services, specifically regarding the consumption of diversified foods during pregnancy [11]. These barriers

encompass heavy workloads, food taboos and restrictions, inadequate support from husbands, insufficient economic resources, limited awareness and low educational attainment among women, poor dietary practices, and a lack of physical access to health facilities [11]. Research conducted in low-and-middle income countries has also identified numerous barriers to adequate heathcare during pregnancy [12]. Key nutritional challenges include low income, insufficient knowledge regarding nutritious dietary practices, limited access to a variety of foods, seasonal fluctuations in food availability, religious fasting, intentional food restrictions aimed at controlling fetal size, and the consumption of alcohol [10].

A study conducted in a peri-urban area of a Tanzanian city indicated that the food choices of PLHIV and their families are influenced by a range of factors. These include the sensory properties of food, the family's knowledge of nutrition, societal stigma, dietary preferences, and other relevant elements [13]. Mmari and colleagues conducted a study among pregnant women in the Kagera region of Tanzania and discovered that these women are often restricted from eating senene, a protein-rich food. This restriction is based on the belief that consuming senene could result in giving birth to children with cone-shaped heads, similar to that of the senene, or children with bald heads [14]. Kavle and Landry documented that barriers to achieving an optimal maternal diet among pregnant women living with HIV included inadequate advice from influential family members, cultural beliefs, fear of delivering large babies, economic constraints, food aversions, and poor intra-household food allocation [9]. Effective policy-making and nutritional programming require a thorough understanding of the challenges that pregnant women living with HIV face in accessing nutritious diets. This is especially important in Njombe, a key food-producing region in the country that is still grappling with malnutrition.

Therefore, the present study aimed to explore barriers on accessing a healthy diet among pregnant women living with HIV in the Njombe region of Tanzania. Understanding barriers to access healthy diets among pregnant women living with HIV will provide basis information on design and develop effective social behavioural interventions to improve nutritional patterns among pregnant women living with HIV in Njombe region.

## Methods

### Ethics statement

This study secured ethical approval from the Mbeya Medical Research and Ethics Review Committee, referenced under number SZEC-2439/R.A/V.1/189. Additionally, permission was obtained to conduct research at the regional, district, and health facility levels. All participants provided written informed consent prior to their involvement in the study, and no financial compensation was provided for participation. Data collected during the study were anonymized and will be destroyed upon the project's completion. The recruitment period for the study started from November 25, 2023, to July 9, 2024.

### Study design

A qualitative study employing ethnographic methods was conducted to gather information through Key Informant Interviews (KIIs), In-depth Interviews (IDIs) and Focus Group Discussions (FGDs). Semi-structured interview questions were utilized to capture participants' opinions and experiences regarding the various factors that impede their access to healthy diets, as outlined in the socio-ecological model is appended as S1 Fig.

### Study population

The study population was pregnant women living with HIV aged 25–35 years who were receiving care at the Care and Treatment Clinics (CTC) within selected Health facilities located in Njombe Town Council, Ludewa District Council, Makete District Council, and Makambako Town Council in the Njombe region of Tanzania. The study also engaged a diverse range of stakeholders, including Community Health Workers (CHWs), District Medical Officers (DMOs), District Nutrition Officers (DNuOs), and District Reproductive and Child Health Officers (DRCHOs). Additionally, it involved religious leaders, health care providers responsible for Reproductive and Child Health at the facility level, Council HIV and AIDS Coordinators, as well as local community leaders.

## Study setting

The study was conducted in the Njombe region of mainland Tanzania, situated within the Southern Highlands Zone. The area exhibits the highest HIV prevalence in the country, recorded at 12.7% among adults aged 15 and above, compared to a national prevalence of 4.4% for adults [15]. The research team deliberately selected two rural councils and two urban councils to obtain more targeted information specific to the area. Health facilities were chosen based on the number of people living with HIVwho access CTC, with a particular emphasis on facilities reporting a high prevalence of cases. The selected councils comprised Ludewa District Council and Makete District Council, representing the rural councils, and Njombe Town Council and Makambako Town Council, representing the urban councils. The selection of these councils and the age range of 25–35 years old was informed by the THIS 2022–2023 [15].

## Selection procedures and data collection

A purposive sampling strategy was implemented to select participants for Focus Group Discussions (FGDs), In-Depth Interviews (IDI) and Key Informant Interviews (KII), to elicit comprehensive insights relevant to the study questions. The research team selected key informants based on their roles in influencing HIV pregnant women or advising on dietary practices and norms. The participants for the FGDs were chosen for their homogeneous socio-demographic character- istics. This approach ensures that commonalities, such as age range, foster an environment conducive to open dialogue and discussion, as opposed to incorporating a diverse age spectrum. The FGDs interview guide is appended as S1 Text. Participants for the IDI and KII included individuals with expertise in nutrition and maternal health initiatives, as well as recognized leaders within the selected community.

The DNuOs facilitated the selection of eligible participants for KIIs and identified religious leaders for IDIs. CHWs played a crucial role in selecting pregnant women living with HIV, as well as women and men for FGDs, utilizing their understanding of local families. Local leaders were tasked with securing suitable locations for conducting interviews and discussions. The KII interview guide is appexed as S2 Text.

A total of 12 FGD sessions were conducted, which included 4 sessions for pregnant women living with HIV, 4 for women of reproductive age, and 4 for men. The overall number of participants in the FGDs for this study was 120, com- prising 33 pregnant women living with HIV, 47 women of reproductive age, and 40 men. Tables 1 provides a summary of the study's FGD. Each FGD session had between 7 and 12 participants and was held in a suitable environment for the respondents. Table 2 provides summary for IDI and KII, 56 interviews were carried out across the study region. Of 56 interviews, 22 were conducted in urban and 24 in rural areas.

## Data analysis

Verbatim transcriptions of the FGDs, IDI and KIIs were carefully conducted by research assistants and notetakers. To ensure the highest level of accuracy, qualified qualitative personnel undertook a thorough verification process by

**Table 1. Distribution of FGDs Conducted by category and study site.**

| Study site | FGD sessions and categories [7–12 participants in each session] | | | Total sessions |
|---|---|---|---|---|
| | Pregnant women living with HIV | Women (25–35 years) | Men (25–49 years) | |
| Ludewa District Council [rural] | 1 | 1 | 1 | 3 |
| Njombe municipality [urban] | 1 | 1 | 1 | 3 |
| Makete [rural] | 1 | 1 | 1 | 3 |
| Makambako [urban] | 1 | 1 | 1 | 3 |
| Total number of FGD sessions | 4 | 4 | 4 | 12 |
| Total number of participants | 33 | 47 | 40 | 120 |

**Table 2. Distribution of the number of IDIs and KIIs conducted.**

| Respondents | IDI | KII |
|---|---|---|
| Pregnant women living with HIV | 8 | – |
| Male partner | 8 | – |
| Community health worker | 4 | – |
| Religious leader | 4 | – |
| District Medical Officers | – | 4 |
| District Nutrition Officer | – | 4 |
| District Reproductive and Child Health Coordinator | – | 4 |
| Community HIV coordinator | – | 4 |
| Reproductive and Child Health Incharge at Health Facility | – | 4 |
| **Total** | **24** | **20** |

comparing the transcriptions with the original audio recordings. The interviews were conducted in the Kiswahili language, employing simultaneous transcription and translation techniques to preserve the authenticity of the data collected. Following transcription, the texts underwent a rigorous thematic coding process for comprehensive content analysis and interpretation. The research team, guided by individuals with expertise in qualitative methodologies, engaged in multiple readings of the texts to formulate detailed codes is appexed as S3 Text. Additionally, a trained researcher coded a subset of the transcripts using the developed codebook to assess the quality and reliability of the data for interpretation and eventual application. The codebook, developed iteratively after the completion of the transcription process. The themes identified were collaboratively discussed and agreed upon by all research team members, enriching the analysis. This thematic analysis involved systematically categorizing data into relevant codes, in alignment with the study's objectives, thereby enhancing the credibility and validity of the findings obtained using MAXQDA software.

## Results

This section analyzes the various barriers that impede access to healthy diets for this vulnerable population. It highlights the complexities that must be addressed to ensure equitable access to nutritious food. The findings were classified into four domains using socio- ecological perspective [16,17].

### Individual factors

**Knowledge of nutritious diets.** Many pregnant women living with HIV in the Njombe region have a limited understanding of what constitutes a nutritious diet. Their food choices are often based on what is readily available in their environment, which may not provide a diverse selection that meets the broader criteria of a balanced diet.

*"… I eat three times. I eat in the morning before going to the farm, I carry Ugali or Kande to the farm for the afternoon, and then I eat again in the evening when I am back home. The foods we eat most are Ugali, potatoes, and Makande"* (HIV pregnant woman, IDI, 1/12/2023, Usagatikwa, Makete).

Another participant said

*"A balanced meal is food like rice, ugali, bananas, and potatoes" (FGD WOMEN, Respondent No. 6, 06/12/2023, Makambako).*

**Unhealthy behaviors regarding nutritious diets.** The study investigated the phenomenon of cravings among pregnant women, revealing that approximately 25% of respondents reported experiencing geophagy. Geophagy is defined

Global Public Health

as a craving for non-nutritive items, such as clay soil, charcoal, and uncooked rice or maize flour. Such cravings can lead to the displacement of more nutrient-rich foods in the diet, which may have adverse implications for the overall health of pregnant women living with HIV.

*"We are not allowed to eat clay but the baby in the womb demands it"* (Respondent #1, Pregnant Women FGD, 30/11/2023, Mlangali Ludewa)

**Financial constraints.** Keeping a nutritious diet can be difficult, especially when financial limitations restrict access to a variety of foods.

*"It's very difficult to provide my wife with everything she needs for her nutrition because we can't afford it. We grow maize, but we can't eat Ugali every day. To buy meat or fish, we need to sell maize to get money, but one bag of maize costs less than a kilogram of meat. Additionally, to sell that bag of maize, I must go to the main market in the nearby village, which costs me about TZS 4000 for the motorbike fee"* (Respondent unknown, Men FGD, 02/12/2023, Tandala, Makete).

Another participant said

*"I would have to choose between buying one egg for TZS 500 when I know that with TZS 500, I could buy a meal for my whole family"* (HIV pregnant woman, IDI, 5/12/2023, Kivavi, Makambako)

### Community factors

These community factors include sociocultural norms, dietary restrictions, alcohol consumption, stigmatization, the influence of spouses, and the role of community health workers in promoting healthy eating habits.

**Sociocultural norms.** Participants reported that pregnant women were advised against eating eggs to prevent their babies from being born without hair.

*"According to our traditions, they avoid eating boiled eggs while pregnant because the child is born without hair"* (Respondent # 3, HIV Pregnant women, FGD, 30/11/2023, Mlangali, Ludewa).

In another part of the study area, after a decade of extensive maternal health education, women can now eat eggs freely because healthcare facilities are equipped with advanced technology and skilled personnel to manage any complications.

*"Currently, we do not have any customs or traditions that forbid certain intake of food. There is also no religion that prohibits that either; pregnant women eat whatever they feel like"* (IDI, 7/12/2023, Idundilanga, Njombe)

**Food restrictions and alcohol consumption.** In Ludewa, pregnant women are strictly prohibited from consuming alcohol. If a pregnant woman is found drinking at a local bar, she may face a fine of up to TZS 50,000 from the local authorities.

*"I mean staying out late at night... That's why even our village government across all the villages in the Ludewa district, provide guidelines for chairpersons and local leaders that pregnant women should not stay out late at local bars or places of entertainment beyond midnight. If seen outside at that time, she is arrested, taken to the appropriate place, and may be fined and face other punishments"* (Respondent #12, FGD, 2/12/2023, Mlangali, Ludewa).

Despite the restrictions, some pregnant women who were used to drinking heavily before pregnancy find it difficult to stop once they become pregnant. Even with the fines and guidelines imposed, they continue to drink.

*"But in our societies, you will still find a pregnant woman who drinks alcohol, sometimes starting from morning until evening, and as a result, fails to prepare and eat proper basic meals because of too much drinking"* (Religious leader, IDI, 02/12/2023, Mlangali, Ludewa)

**Stigmatization.** Most respondents noted that stigmatization has significantly decreased, attributing this change to widespread awareness campaigns about HIV/AIDS. Previously, there were misconceptions in the community that the disease could be transmitted through casual contact, such as shaking hands or sharing meals.

*"Honestly, the issue of stigma here has completely ended. I am HIV affected, and I eat and talk with my colleagues without any issue; they understand me. When you go to the office with any kind of problem, they understand you. In other words, stigma here has completely disappeared"* (Respondent #11, Men, FGD, 2/12/2023, Tandala, Makete)

In Makambako District Council, pregnant women living with HIV from Kivavi still claim on the pouches of stigmatization at individual and family levels.

*"When I go to the care and treatment clinics (CTC) for services, if I happen to see my neighbor, I turn around and pretend to head to a different room until my neighbor leaves, and then I return to the CTC room. It's best when they don't know because if they find out, they would speak badly about you and your family. The stigma is still very high"* (HIV pregnant woman, FGD, 5/12/2023, Kivavi, Makambako)

**The influence and support of spouses.** This encompasses emotional care when the woman feels unwell due to pregnancy, providing the food that the pregnant woman craves or needs, accompanying her to clinic visits, assisting with domestic and farming tasks, as well as offering financial assistance for all delivery-associated expenses.

*"Pregnant women, especially those living with HIV, find great comfort from nurses and doctors, but not from the community. Out of 100, your community offers support only about 25% of the time. Once you're pregnant, that becomes your burden because from the moment the pregnancy is conceived, you'll be raising it in difficult circumstances until the child is born"* (Respondent #14, Pregnant women, FGD, 30/11/2023, Tandala, Makete).

Many women continued to work during their pregnancies, with some receiving support from their partners starting around the seventh month.

*"Most women in this area prefer to work, often continuing to go to the farm until they are seven or eight months pregnant. Typically, they only start to rest as they approach the final months. It is during this time that their spouses provide the most help"* (DRCHO, KII, 6/12/2023, MjiMwema, Njombe)

## Organizational factors

The factors within organizations that can either facilitate or obstruct access to a nutritious diet for pregnant women living with HIV include nutritional education, the knowledge and skills of healthcare providers, the availability of healthcare professionals, availability of working tools and the proximity to healthcare services.

**Nutritional education.** Many participants noted the availability of various nutritional programs and events, organized in collaboration with other stakeholders. The aim is to provide the community with nutrition education, training materials and, equipment.

*"Nutritional education is often provided by healthcare providers who train community health workers to go and offer counseling services to the community. We provide nutritional education at community meetings and events, and we also take turns attending radio programs"* (KII, 5/12/2023, Makambako)

**Knowledge and skills of healthcare providers.** District and ward healthcare providers work together with various community stakeholders, including social welfare committees, community health workers, Mother Champions, and Lishe clubs, to promote nutrition education. However, these providers have not received sufficient refresher training for an extended period.

*"Knowledge capacity varies because we have new employees and some of whom have not received training. So, it is true that there are those with great ability and provide education well because of the skills they have, and there are others who still face challenges. We are still building their capacity on how to provide education in the community and supervise primary health care workers; this is an ongoing work"* (KII, 6/12/2023, Mji Mwema, Njombe)

**Availability of healthcare providers.** Most villages have dispensaries, but they usually employ only two service providers each.

*"The distance to the healthcare center is not far, but upon arrival, you may spend an entire day there due to the long que. Sometimes there is only one nurse available, but most times only a nurse and a doctor are present"* (Pregnant Woman, FGD, 4/12/3023, Njombe Mjini, Njombe)

**Availability of working tools.** Most of health facilities have essential anthropometric assessment tools; however, the new constructed facilities experienced inadequaency of equipments, guidelines, jobs aids and other training

*"In my council, we have about nine new health facilities. They are having a bit of a challenge getting a weighing scale, but the rest of the equipment is there and in use"* (KII, 4/12/2023 Iwama, Makete)

**Proximity to healthcare services.** We examined the accessibility of healthcare services for pregnant women and found that all the study districts have a sufficient number of facilities located near their villages. For instance, Ludewa has 77 villages and 78 facilities, while Makambako has 25 villages with an equal number of facilities. This suggests that each village is served by its own healthcare facility, which minimizes commuting challenges for residents.

*"Many people come from one or two kilometers away, which is not far, and transportation is available. If it's about three or two kilometers, you can take a bajaji, and even if you don't take a bajaji, and you can manage you can even walk and arrive at the station on time"* (HIV Pregnant woman, IDI, 5/12/2023, Njombe Mjini)

## Environmental/contextual factors

The study also examines key environmental factors, including food seasonality and availability in the study districts, and how these factors affect the accessibility of a healthy diet for pregnant women living with HIV.

**Food availability and seasonability.**  Respondents noted that there is an ample supply of food during the rainy season. Even after the season ends, the use of irrigation farming enables continuous food production throughout the year. Locally grown foods, including maize, Irish potatoes, beans, bananas, oranges, avocados, mangoes, and pineapples, as well as vegetables such as potato leaves, Chinese cabbage, radish leaves, spinach, and pumpkin leaves, are affordable and accessible to most people.

"*For instance, in Makambako we grow maize, potatoes and bananas. We also have fruits such as avocado, oranges, mangoes, pineaples and a lot of vegetables which are grown near the valleys*" (Men, IDI, 6/12/2023, Makambako)

Interestingly, the region lacks a strong culture of animal husbandry. Only a small portion of the population raises chickens and Simbilisi (guinea pigs), which provides some access to eggs and this particular type of meat. The breeding of other domestic animals, such as cows and goats, is rare, leading to limited availability and high prices for milk, cow meat, and goat meat for most of the population.

"*In some areas here in Ludewa, we have very few livestock, making animal keeping uncommon. In some villages they only eat meat during Christmas, Easter, Eid, and special events only, people go extended periods of time without seeing meat*" (KII, 1/12/2023, Ludewa)

## Discussion

This study employed socio-ecological model to explore the various obstacles pregnant women living with HIV encounter in accessing healthy diets in Tanzania's Njombe region [16,17]. It offers a detailed analysis of these challenges across individual, community, environmental, and organisational levels. Characterised by its fertile soil and favourable climate, Njombe region is well-suited for agriculture, providing a conducive environment for increasing the production of nutritious foods. Despite being recognised as a food basket, Njombe struggles with malnutrition, highlighting the urgent need to explore factors hindering access to healthy diet among this group which has increased nutrition needs.

The study identified several factors that impede pregnant women living with HIV in the Njombe region from sustaining healthy diets. These barriers include inadequate support from spouses and family members, a shortage of healthcare providers, limited knowledge of nutrition, the excessive consumption of alcohol among pregnant women, and excessive workloads and responsibilities, income constraints [1,2,4,5]. Pregnant women lacking adequate emotional and practical support from their spouses and families, along with those overwhelmed by household chores, are at an increased risk of experiencing exhaustion, stress, and diminished appetite. Such conditions can lead to weakened immunity, adversely affecting both maternal health and the health of the unborn child [1,2,4,5]. Limited understanding of nutrition constitutes a significant obstacle, as it is challenging to implement effective dietary practices without a thorough grasp of nutritional principles. While the issue of stigmatization was largely mitigated at the community level, it remains a concern at the individual and family levels, especially among serodiscordant couples. Additionally, consumption of alcohol by pregnant women is still prevalent in certain areas, posing a threat to their overall health maintenance. Finally, the scarcity of healthcare providers may result in insufficient healthcare services, particularly when the demand for such services surpasses the available resources [1,2,4,5]. The information presented here is based solely on the participants' responses and is specific to the areas covered. Although we aimed to gather information on barriers to accessing healthy diets among pregnant women living with HIV, there may be overreported or overlooked information. So, this study should be understood as applicable only to regions with similar socio economic and cultural contexts as Njome region of Tanzania. Furthermore, the study did not capture information on food intake thus misses the opportunity to deeply understand the consumption pattern particularly on micronutrients –rich foods.

Our study indicates that pregnant women in the Njombe region possess a basic understanding of the importance of maintaining a nutritious diet during pregnancy. However, they struggle to adhere to healthy eating practices due to various barriers that hinder their ability to apply nutrition information. A related study conducted in the Makete district by Mruma & Mkhai revealed that pregnant women lacked knowledge regarding proper meal planning and preparation to support their health during pregnancy [18]. This situation has resulted in limited dietary diversity, as the majority of women rely on homegrown foods, which has led to a predominantly plant-based diet. Similar dietary patterns have been documented in other low-income countries [19,20]. Achieving a healthy diet involves a combination of intrinsic personal factors and external influences related to environmental and systemic conditions. The motivation to adopt healthy dietary practices is primarily driven by effective nutrition education, as well as a maternal instinct to safeguard the health of the unborn child [21]. The availability and accessibility of a diverse range of nutritious food options are vital for enhancing the health outcomes of pregnant women living with HIV and their unborn children. Inadequate dietary intake during pregnancy can lead to maternal malnutrition, which may, in turn, have enduring detrimental effects on the health of the child [22]. A study conducted in Spain indicates that it is advisable for pregnant women to adopt a balanced diet. This diet should consist of 3–4 servings of dairy products, 2–3 servings of meat, fish, or eggs, 3 servings of fruit, 4–5 servings of vegetables or greens, and 7–8 servings of cereals or legumes on a daily basis [23]. Adhering to these dietary recommendations can contribute to the overall health and well-being of both the mother and the developing fetus [23]. In the Njombe region, the dietary patterns of pregnant women are characterized by limited variety, predominantly consisting of starch staples along-side a select range of vegetables. There is a notable limited intake of dairy products, meat, fish, eggs, and fruits among the majority of this population group.

The research highlighted the importance of community support for pregnant women, emphasizing that the gestation period is particularly critical for both the expectant mother and the fetus. During pregnancy, it is essential to ensure the well-being of both the mother and the child. Evidence indicates that women who receive strong support from their hus-bands at the household level are more likely to consume iron and folic acid, calcium supplements, and a varied diet compared to those who receive little support [10]. A supportive spouse is more likely to assist their partner with household responsibilities, similar to how a supportive network of friends and family can aid a pregnant woman in managing her community obligations. This type of assistance plays a critical role in alleviating pregnancy-related risks, including fatigue, stress, miscarriages, reduced immunity, and the potential for illnesses. Research indicates that an excessive workload during pregnancy can have a significant impact on fetal development, particularly concerning low birth weight (LBW), being small for gestational age (SGA), and the likelihood of premature birth [24]. A recent study has indicated similar effects, including miscarriage, hypertension, pre-eclampsia, and various obstetric complications [25].

The current study identified a pattern of alcohol consumption among pregnant women, along with cravings that hinder their ability to consume nutritious foods. These cravings included a desire for clay soil, unripe sour fruits, and raw foods. Evidence indicates that these food aversions and cravings serve as significant physiological obstacles to achieving opti-mal dietary intake and practices [26]. Cravings, referred to as pica, can restrict a woman's intake of essential nutrients. One specific type of pica, known as geophagy, involves the consumption of clay-like substances. This practice can bind to important micronutrients such as zinc and iron, potentially resulting in nutrient deficiencies. Research has shown a connection between geophagy and iron-deficiency anaemia [27]. Similar misconceptions exist in other African countries. Studies conducted in Ethiopia [28], Congo [29], and Uganda [30] have highlighted how these traditional beliefs have pre-vented pregnant women from consuming a balanced diet for many years.

Income constraints have significantly affected individuals' ability to consume a healthy diet. Many people find it unaf-fordable to include the recommended proportions of dairy products, meat, fish, eggs, and fruits due to their low purchasing power. In the Njombe region, most respondents depend on agriculture primarily for domestic consumption rather than commercial purposes. Given their relatively limited income, this poses a significant challenge for pregnant women trying to adopt a healthy dietary behaviour. Comparable results have been observed in several countries, including India [10],

PLOS Global Public Health

Indonesia [21], Pakistan [26], as well as in Burkina Faso, Kenya, and Tanzania [27], and Niger [31]. The study explored the challenges encountered by pregnant women living with HIV in the Njombe region regarding their access to a healthy diet. It assessed various factors that may contribute to difficulties in securing proper nutrition while identifying certain elements that exert minimal influence. Significantly, factors such as food availability, religious and socio-cultural norms, stigmatization at the community and family levels, proximity to healthcare services, and substance use were not found to substantially hinder access to healthy diets. Additionally, the community has largely moved away from harmful traditional practices, and healthcare services are reasonably accessible to the population.

Understanding the barriers to accessing healthy diets among pregnant women living with HIV is crucial for the development of effective social and behavioral interventions aimed at enhancing nutritional practices within this demographic in the Njombe region of Tanzania. This matter is particularly significant, as inadequate nutrition has profound implications for economic stability, societal health, and the development of future generations and communities. Tanzania has made substantial investments across various sectors, resulting in a significant transformation in nutrition. To harness this opportunity and ensure it leads to improved population health, enhanced nutritional status, greater food security, and overall well-being, this study identifies critical barriers to healthy diets faced by pregnant women living with HIV. By making these challenges visible to both government and non-government stakeholders, we can drive the creation of a robust and comprehensive framework of policies, strategies, and guidelines that will empower these vulnerable communities and promote healthier outcomes.

In conclusion, the study unequivocally identified substantial barriers that prevent pregnant women living with HIV in the Njombe region of Tanzania from accessing healthy diets. Promoting nutritional awareness within these communities is imperative for enhancing the knowledge, skills, and motivation of pregnant women, their partners, and the broader community to adopt healthy dietary behaviours. While various barriers to a healthy diet may exist, motivation and intentionality in pursuing such dietary choices are equally crucial. Even in the face of challenges, individuals with a robust understanding of nutrition are more likely to identify alternative strategies to maintain healthy diets. Moreover, communities in Njombe must adopt a more diverse approach to animal breeding, shifting their focus to include cattle and goats alongside the commonly practiced breeding of chickens and guinea pigs. Expanding these animal breeding initiatives will significantly enhance the availability of protein-rich foods, thereby ensuring better access to and affordability of essential protein-based foods.

## Supporting information

**S1 Fig. Socio-ecological model shows barriers to accessing healthy diets.**
(TIF)

**S1 Text. FGDs interview guide.**
(DOC)

**S2 Text. KII guide.**
(DOC)

**S3 Text. Thematic and key factors.**
(DOC)

**S1 Checklist. STROBE Checklist.**
(DOC)

## Author contributions

**Conceptualization:** Julieth Shine, Vera Kwara, Deborah Esau, Fatma Abdallah, Anna Zangira, Aika Lekey, Abela Twinomujuni, Maria Ngilisho, Zahara Amiri, Germana H. Leyna, Ray Mrisho Masumo.

**Data curation:** Abela Twinomujuni.

**Formal analysis:** Julieth Shine, Fatma Abdallah, Elizabeth Lyimo, Winfrida Akyoo, Germana H. Leyna, Ray Mrisho Masumo.

**Funding acquisition:** Vera Kwara, Deborah Esau.

**Investigation:** Maria Ngilisho.

**Methodology:** Julieth Shine, Anna Zangira, Aika Lekey, Zahara Amiri, Elizabeth Lyimo, Winfrida Akyoo, Ray Mrisho Masumo.

**Project administration:** Vera Kwara, Deborah Esau, Fatma Abdallah, Aika Lekey, Germana H. Leyna.

**Supervision:** Julieth Shine, Vera Kwara, Deborah Esau, Germana H. Leyna, Ray Mrisho Masumo.

**Validation:** Abela Twinomujuni, Winfrida Akyoo, Ray Mrisho Masumo.

**Writing – original draft:** Julieth Shine, Vera Kwara, Deborah Esau, Fatma Abdallah, Anna Zangira, Aika Lekey, Abela Twinomujuni, Maria Ngilisho, Zahara Amiri, Elizabeth Lyimo, Winfrida Akyoo, Germana H. Leyna, Ray Mrisho Masumo.

**Writing – review & editing:** Julieth Shine, Vera Kwara, Deborah Esau, Fatma Abdallah, Anna Zangira, Aika Lekey, Abela Twinomujuni, Maria Ngilisho, Zahara Amiri, Elizabeth Lyimo, Winfrida Akyoo, Germana H. Leyna, Ray Mrisho Masumo.

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
