## [Decision Letter · Decision Letter 0]

29 May 2025

PGPH-D-25-00561

Barriers to accessing a healthy diet among HIV pregnant women in the Njombe region of Tanzania: A qualitative study

Dear Dr. Masumo,

Thank you for submitting your manuscript to PLOS Global Public Health. After careful consideration, we feel that it has merit but does not fully meet PLOS Global Public Health’s publication criteria as it currently stands. Therefore, we invite you to submit a revised version of the manuscript that addresses the points raised during the review process.

Three reviewers have assessed your manuscript. Their comments are appended below, and in the attached file. Please carefully review their suggestions and revise your manuscript accordingly, providing a detailed point-by-point response to the reviewers. 

We look forward to receiving your revised manuscript.

Kind regards,

Sarah Jose, Ph.D.

Staff Editor

Journal Requirements:

1. We have noticed that you have uploaded Supporting Information files, but you have not included a list of legends. Please add a full list of legends for your Supporting Information files after the references list.

Additional Editor Comments (if provided):

Reviewers' comments:

Reviewer's Responses to Questions

**Comments to the Author**

1. Does this manuscript meet PLOS Global Public Health’s publication criteria ? Is the manuscript technically sound, and do the data support the conclusions? The manuscript must describe methodologically and ethically rigorous research with conclusions that are appropriately drawn based on the data presented.

Reviewer #1: Yes

Reviewer #2: Partly

Reviewer #3: Yes

2. Has the statistical analysis been performed appropriately and rigorously?

Reviewer #1: N/A

Reviewer #2: N/A

Reviewer #3: Yes

3. Have the authors made all data underlying the findings in their manuscript fully available (please refer to the Data Availability Statement at the start of the manuscript PDF file)?

Reviewer #1: Yes

Reviewer #2: Yes

Reviewer #3: Yes

4. Is the manuscript presented in an intelligible fashion and written in standard English?

Reviewer #1: Yes

Reviewer #2: Yes

Reviewer #3: Yes

5. Review Comments to the Author

Reviewer #1: Overall, the authors should provide a more thorough justification for their study, particularly explaining why the Njombe region was selected as the study area. Additionally, the choice of the study population—pregnant women living with HIV aged 25-35 years—requires justification, as the typical reproductive age range is 15-49 years. This can be addressed by offering a clear description of the inclusion and exclusion criteria used. While the study focuses on barriers, it would benefit from also exploring enablers, as this would provide a more balanced perspective and broaden the study’s applicability. I have uploaded a file with my detailed review feedback for the authors.

Reviewer #2: The authors conducted a well-designed qualitative study that explored the barriers to accessing a healthy diet among HIV-positive pregnant women in the Njombe region of Tanzania. This study provides important context-specific insights into the unique nutritional challenges faced by HIV-infected pregnant women, a population often underrepresented in existing nutritional research literature. It has the potential to add great value to the understanding of individual, community, organizational, and environmental barriers. Overall, this paper offers valuable implications for maternal health interventions and policy development. However, there are some major concerns for the authors to consider:

1. As mentioned in the Introduction, existing studies have identified several barriers such as influence of family members, economic limitations, and food scarcity. It is yet not clear how this study uniquely contributes to filling a particular gap in the literature.

2. I encourage the authors to further unpack what factors mattered more across different groups (e.g., do HIV pregnant women, women of reproductive age, and men had similar or different perceptions on the topics).

3. I suggest writing out the exact number of participants in each category in Table 1. It is confusing right now as it only shows the number of FGD sessions (which is 1 in all categories and not very informative).

4. The Results and Discussion sections were not very well structured. I appreciate the authors citing quotes from the participants but currently It is difficult to tell which factors were considered to be more important (or more often brought up) by which group of participants.

5. Although the study is frameworked as a qualitative research, it would be still valuable to have summary tables in the Results section to provide counts of different thematic themes and key factors.

6. The Discussion was comprehensive, but it could be more directly tied back to the initial objectives and the research questions. Drawing out clear links between the findings and the broader context of maternal and HIV nutrition policies in Tanzania could make this piece more effectively demonstrate the implications and practical relevance of your results.

7. Related to the comment above, the Discussion section could benefit from an emphasis on implications for policy and practice.

Reviewer #3: The study is of significant relevance, as it addresses the barriers to healthy diets among HIV-positive pregnant women in Tanzania at a critical juncture. The application of ethnographic qualitative methods, including focus group discussions, in-depth interviews, and key informant interviews, with a diverse range of participants such as HIV-positive women, men, healthcare providers, and community leaders enhances the robustness of the study. Employing the socio-ecological model to categorize findings into individual, community, organizational, and environmental factors is executed proficiently. The connection between the findings and existing literature is effectively articulated, and the practical implications are clearly highlighted. Furthermore, the transparent declaration of ethical considerations, data availability, funding sources, and competing interests contributes positively to the overall integrity of the research. The author can try to reduce the words in the abstract as it is a bit too detailed. For PLOS Global Public Health, abstracts should be sharp and within the word limit (preferably under 300 words). Try to reduce it by focusing on major findings, key methods and conclusion. Generally, it is well researched pare and excellent work for the author

6. PLOS authors have the option to publish the peer review history of their article (what does this mean? ). If published, this will include your full peer review and any attached files.

**Do you want your identity to be public for this peer review?** For information about this choice, including consent withdrawal, please see our Privacy Policy .

Reviewer #1: **Yes: ** Amon Exavery, PhD

Reviewer #2: No

Reviewer #3: **Yes: ** Deborah Tembo

---

## [Decision Letter · Decision Letter 1]

7 Aug 2025

PGPH-D-25-00561R1

Exploring barriers to accessing healthy diets among pregnant women living with HIV in the Njombe region, Tanzania: A qualitative study

Dear Dr. Masumo,

Thank you for submitting your manuscript to PLOS Global Public Health. After careful consideration, we feel that it has merit but does not fully meet PLOS Global Public Health’s publication criteria as it currently stands. Therefore, we invite you to submit a revised version of the manuscript that addresses the points raised during the review process.

We look forward to receiving your revised manuscript.

Kind regards,

Miho Sato

Academic Editor

Journal Requirements:

Additional Editor Comments (if provided):

Following careful consideration of the revised manuscript and the second-round evaluations, I am pleased to inform you that the manuscript is close to being accepted. However, we request that you address a few remaining concerns before a final decision can be made. Please kindly follow the additional comments made by the Reviewer.

Reviewers' comments:

Reviewer's Responses to Questions

**Comments to the Author**

1. If the authors have adequately addressed your comments raised in a previous round of review and you feel that this manuscript is now acceptable for publication, you may indicate that here to bypass the “Comments to the Author” section, enter your conflict of interest statement in the “Confidential to Editor” section, and submit your "Accept" recommendation.

Reviewer #1: All comments have been addressed

Reviewer #2: All comments have been addressed

Reviewer #3: All comments have been addressed

2. Does this manuscript meet PLOS Global Public Health’s publication criteria ? Is the manuscript technically sound, and do the data support the conclusions? The manuscript must describe methodologically and ethically rigorous research with conclusions that are appropriately drawn based on the data presented.

Reviewer #1: Yes

Reviewer #2: Yes

Reviewer #3: Partly

3. Has the statistical analysis been performed appropriately and rigorously?

Reviewer #1: N/A

Reviewer #2: N/A

Reviewer #3: Yes

4. Have the authors made all data underlying the findings in their manuscript fully available (please refer to the Data Availability Statement at the start of the manuscript PDF file)?

Reviewer #1: Yes

Reviewer #2: Yes

Reviewer #3: Yes

5. Is the manuscript presented in an intelligible fashion and written in standard English?

Reviewer #1: Yes

Reviewer #2: Yes

Reviewer #3: Yes

6. Review Comments to the Author

Reviewer #1: The authors have addressed my previous comments, and I commend them for their efforts. I have no further comments, except for one minor observation: the word "appexed" appears to be a typographical error and should likely be corrected to "appended".

Reviewer #2: All comments have been addressed. I appreciate the authors adding the important content in discussion as well as appendix table.

Reviewer #3: This manuscript examines a critical public health issue: the barriers to healthy diets faced by pregnant women living with HIV in a high-burden region of Tanzania. The study is timely, relevant, and thoroughly grounded in both theoretical and contextual frameworks. Using the socio-ecological model and ethnographic methods helps develop a rich understanding of the complex barriers operating at individual, community, organizational, and environmental levels.

However, the manuscript can be improved through clearer language, deeper analysis in the discussion, and a stronger connection between the results and existing literature. These improvements will increase the manuscript’s impact and readability for a global audience

7. PLOS authors have the option to publish the peer review history of their article (what does this mean? ). If published, this will include your full peer review and any attached files.

**Do you want your identity to be public for this peer review?** For information about this choice, including consent withdrawal, please see our Privacy Policy .

Reviewer #1: **Yes: ** Amon Exavery, PhD

Reviewer #2: No

Reviewer #3: **Yes: ** Deborah Tembo

---

## [Editor Report · Decision Letter 2]

17 Sep 2025

Exploring barriers to accessing healthy diets among pregnant women living with HIV in the Njombe region, Tanzania: A qualitative study

PGPH-D-25-00561R2

Dear Dr. Masumo,

We are pleased to inform you that your manuscript 'Exploring barriers to accessing healthy diets among pregnant women living with HIV in the Njombe region, Tanzania: A qualitative study' has been provisionally accepted for publication in PLOS Global Public Health.

Best regards,

Miho Sato

Academic Editor